# The Impact of Various Promotional Activities on Ebola Prevention Behaviors and Psychosocial Factors Predicting Ebola Prevention Behaviors in the Gambia Evaluation of Ebola Prevention Promotions

**DOI:** 10.3390/ijerph16112020

**Published:** 2019-06-06

**Authors:** Anna E. Gamma, Jurgita Slekiene, Hans-Joachim Mosler

**Affiliations:** EAWAG, Swiss Federal Institute of Aquatic Science & Technology, Ueberlandstrasse 133, CH-8600 Duebendorf, Switzerland; anna.gamma@gmx.net (A.E.G.); jurgita.slekiene@eawag.ch (J.S.)

**Keywords:** Ebola virus disease (EVD) prevention, behavior change, psychosocial factors, RANAS model, handwashing with soap, emergencies and outbreaks, mediation analysis

## Abstract

The outbreak of the Ebola virus disease (EVD) from 2014 to 2016 is over. However, several outbreaks of contagious diseases have already arisen and will recur. This paper aims to evaluate the effectiveness of EVD prevention promotions in the Gambia and to assess the psychosocial factors that steer three behaviors: handwashing with soap, calling the Ebola Hotline, and not touching a person who might be suffering from EVD. In 2015, data were gathered from 498 primary care providers. The questionnaire was based on psychosocial factors from the risks, attitudes, norms, abilities, and self-regulation (RANAS) model. Three promotional activities were significantly associated with psychosocial factors of handwashing and, thus, with increased handwashing behavior: the home visit, posters, and info sheets. Norm factors, especially the perception of what other people do, had a great impact on handwashing with soap and on calling the Ebola Hotline. The perceived certainty that a behavior will prevent a disease was a predictor for all three protection behaviors. Commitment to the behavior emerged as especially relevant for the intention to call the Ebola Hotline and for not touching a person who might be suffering from EVD. Health behavior change programs should rely on evidence to target the right psychosocial factors and to maximize their effects on prevention behaviors, especially in emergency contexts.

## 1. Introduction

During the previous outbreak of Ebola virus disease (EVD) in West Africa, 28,646 cases were confirmed, probable, or suspected, and 11,323 deaths were reported [1]. The ebola virus disease is a severe illness with a mortality rate between 25% and 90% and an average fatality rate of around 50%. Fruit bats (family Pteropodidae) are considered a reservoir of EVD. They spread the virus to chimpanzees, gorillas, monkeys, and humans. Human-to-human transmission occurs via blood, body fluids, contaminated objects, handling of dead bodies during funerals, and sexual transmission after recovery [2]. Although promising effects of an EVD vaccine have recently been confirmed in Guinea [3], outbreaks of contagious diseases, such as the emerging Zika virus disease, will recur. Besides vaccines against contagious diseases, preventive behaviors play a crucial role in impeding further transmission in a population. The WHO recommends the following package of interventions to control an outbreak in general: surveillance, infection prevention, and control practices, case management, contact tracing, community engagement, social mobilization, safe burials, and a good laboratory service [2]. The spread of EVD was exacerbated and facilitated by weak health systems and the limited capacities of governments to monitor fluid borders [4]. 

Many studies have used knowledges-attitudes-practice (KAP) surveys to assess awareness of the disease. These KAP studies mostly reveal the level of knowledge of the population about Ebola [5,6,7], and some analyze the effects of information sources used [8]. However, evaluations of health interventions in emergency settings are rare [9,10]. Increasing the efficiency of public health interventions requires rigorous evidence about the effectiveness of interventions to change behavioral determinants, behaviors, and their impact on health outcomes [11,12].

In the previous outbreak of EVD in West Africa, which is a disease previously unknown in the affected population, health workers had to address disbeliefs about the disease and strong cultural traditions that contributed to the spread of the virus. These included caring for sick people at home, going to traditional healers, and being in close contact with dead bodies before the burial ceremony. Communication is a key activity during an emergency response [13], but the content of the messages should go beyond simple health information. Awareness-raising and information, both of which were crucial and essential in the affected regions, do not, on their own, necessarily lead to the desired behavior. However, they can build the foundation of a behavioral change over the long term [14,15]. Behaviors are based on processes in the minds of individuals, so the uptake of new protective behaviors requires either that people’s mindsets are in favor of these behaviors or that they change [16]. Therefore, understanding what drives a specific behavior within a specific population or context is essential to developing effective public health interventions, and not only in an epidemic or pandemic [17,18]. Systematic behavioral changes, as proposed by Mosler [18], are based on research from environmental and health psychology. It first systematically assesses the psychosocial factors that steer behavior. Knowledge of the psychosocial factors underlying the desired behaviors can then guide the selection of evidence-based interventions. The final phase of systematic behavior change evaluates the effectiveness of the interventions and the mechanisms of the change [19]. The urgent need for careful evaluation of emergency hygiene promotions has been shown by Contzen and Mosler [20]. They evaluated the effect of various promotional activities on handwashing behavior as a response to a cholera outbreak in Haiti after the earthquake in 2010. The evaluation revealed that several promotional activities had negative associations with behavior, which means people who had experienced the activity reported less handwashing. This finding indicates that the activity might be not only ineffective but even counterproductive. Therefore, accurate evaluations of promotional activities are crucial to maximize their impact and to avoid unwanted effects.

### 1.1. The Current Study

The month before this survey took place in Gambia, the local collaborator, Concern Universal, together with other local partners, implemented four the Ebola prevention promotions: Household visits, posters with information about EVD at public places, EVD information sheets for households, and hygiene kits. The household visits were used to present transmission routes, symptoms of Ebola, and preventive behaviors. The poster displayed the symptoms of Ebola, requests to wash hands with soap and water, and to report signs of Ebola to a health facility. The information sheet contained instructions about signs and symptoms of Ebola, contamination pathways, and prevention measures. The hygiene kits included soap, bleach, material for a tippy tap (www.tippytap.org/the-tippy-tap), cups, a bucket, and a flyer about Ebola. This study aimed to evaluate these Ebola prevention promotions in Gambia. The main objective was to reveal whether the promotions successfully tackled key psychosocial determinants of the prevention behaviors, because this is a precondition for the effectiveness of health promotions and enables understanding of why a promotional activity was effective. Furthermore, it was possible to show which of the key determinants had not been tackled so far by promotional activities. Another objective was to identify the key determinants of the three EVD preventive behaviors of interest: handwashing with soap, calling the Ebola Hotline, and not touching a person who might be suffering from EVD. The findings of this study can be used to improve the EVD response activities that have already been implemented by including the key psychosocial determinants that have not been addressed so far and that are, therefore, promising targets for increasing EVD prevention behaviors.

### 1.2. The RANAS Model of Behavior Change

The risks, attitudes, norms, abilities, and self-regulation (RANAS) approach was developed to predict health behavior in developing countries [18]. It offers an effective instrument for identifying psychosocial factors in the water, sanitation, and hygiene (WASH) and health sectors. It has been applied to answer the research questions of the present study. The RANAS approach also enables the effectiveness of promotional activities to be evaluated by looking at their underlying mechanisms. This is done by analyzing whether the interventions successfully tackled the key behavioral factors or not. The applicability of the approach has been demonstrated in various studies [20,21,22,23]. 

The RANAS model includes five blocks of factors. Risk factors include factual knowledge about the transmission of a disease, methods of prevention, personal consequences, perceived vulnerability, and the perceived severity of contracting a disease. Attitude factors include beliefs about the costs and benefits of a particular behavior and feelings associated with the behavior. Norm factors, such as the perception of what others are doing, others’ disapproval, and personal importance, relate to perceived social influence. Ability factors include people’s confidence in the performance of a particular behavior. Self-regulation factors include the management of conflicting goals, distracting cues and barriers, commitment, and remembering the behavior.

### 1.3. The Preventive Behaviors during an Ebola Outbreak

The preventive behaviors during an EVD outbreak include safe burial, regular handwashing with soap, reporting suspected EVD cases to the National Ebola Hotline or a health facility, and not touching a sick person [2]. Because no cases of EVD occurred in Gambia, calling the Ebola Hotline to report a suspected case and not touching someone who might be suffering from EVD could not be measured directly. Therefore, behavioral intention and behavioral willingness were examined for these behaviors. 

This paper presents cross-sectional study results from an EVD response survey in Gambia and addresses four research questions. 

Which are the crucial psychosocial determinants of handwashing with soap at key times under the threat of EVD?

Which are the crucial psychosocial determinants of the intention to call the Ebola Hotline to report a suspected case of EVD?

Which are the crucial psychosocial determinants of the intention not to touch someone who might be suffering from EVD? 

Which promotional activities affect which psychosocial factors and influence the preventive behaviors through these factors?

## 2. Materials and Methods 

### 2.1. Research Area

The Republic of the Gambia is one of Africa’s smallest countries. It is surrounded by Senegal except for its coastline on the Atlantic Ocean at its western end. The Gambia is divided into five administrative regions and one city. The study area consisted of two regions in which Ebola promotion activities were conducted, which included the West Coast Region, comprising 19 communities, and the Lower River Region, comprising 22 communities. Data were collected in all these communities. These are areas with large volumes of passenger transport and goods transport from all sides (Senegal to the north and south and Guinea Bissau to the south). The Gambia trades extensively with neighboring countries through markets that involve large volumes of trucks, passenger vehicles, and other travelers crossing its borders in both directions. 

### 2.2. Participants

The sample includes data from 498 respondents. The interviews were conducted with the member of the household who is responsible for the care of the sick. The household selection was based on random-route sampling, according to the protocol defined by Hoffmeyer-Zlotnik [24]. Following this protocol, the interviewers were sent to different places in the community and instructed to include every third household they encountered on their way. Interventions were implemented in all the communities, so most people living there should have experienced the interventions.

### 2.3. Procedure

The study was conducted in the Gambian households in May and June 2015. A quantitative cross-sectional survey was conducted with structured face-to-face interviews using a paper-and-pencil format. Each interview took around one hour and was held in one of the local languages: Jola, Mandinka, or Fula. A team of 10 local health sector employees were recruited as interviewers. They attended five days of intensive training, during which they learned about the study, its goals, and the theoretical background of the questionnaire. The data collectors practiced interview techniques and the translation of the questions into the local languages. Two supervisors and the local collaborator coordinated and corrected the interviews and accompanied the data collectors in the field during the entire period of the data collection. Each data collector conducted the survey in those communities in which he or she was not working as health workers. All study participants provided their written informed consent prior to the interviews. The study received ethical approval from the School of Medicine and its allied Health Sciences Research and Publication Committee at the University of Gambia.

### 2.4. Promotional Activities 

In the months before the survey took place, the local collaborator, Concern Universal, together with other local partners implemented four promotion activities to help prevent an EVD outbreak in Gambia. The respondents were asked if they had experienced the Ebola prevention promotions or not.

### 2.5. Questionnaire and Measures

A structured questionnaire was developed and pre-tested for this study. The questionnaire was based on the psychosocial factors of the RANAS model [18]. Most of the questions were measured using 5-point Likert scales, which were pretested and extensively discussed and trained with the interviewers. The questionnaire covered the following elements: socio-demographic characteristics, psychosocial factors for handwashing with soap, self-reported handwashing frequencies, the intention to follow the prevention instructions (to call the Ebola Hotline and not to touch a person who might be suffering from EVD), and corresponding psychosocial factors, measures of socio-economic status, remembered promotion activities, and attitudes toward them. 

Additionally, frequency of communication about the Ebola Hotline was included because talking frequency is an important determinant of whether a person will change a certain behavior or not [25]. Various studies have confirmed that communication plays an essential role in a health-related behavioral change [26]. 

The questionnaire was tested at the end of the interviewers’ training to verify its applicability. 

### 2.6. Handwashing with Soap at Key Times

To include the data from all respondents, only the handwashing moments after defecation and before eating were used for analysis. The data collectors asked the respondents how often they washed hands after defecation and before eating. Answers were assessed on a 5-point rating scale from (almost) never to (almost) every time. A mean score was built with the two handwashing questions (Cronbach’s alpha α = 0.75).

### 2.7. Intention to Follow Prevention Instructions

The intention to follow EVD prevention instructions, reporting a suspected EVD case to the Ebola Hotline and not touching sick people, was operationalized through behavioral intention and behavioral willingness. Two direct questions were asked using self-reported 5-point Likert scales, from 1 (not at all) to 5 (very strongly) for calling the Ebola Hotline and from 1 (not at all willing) to 5 (very willing) for not touching sick people (Cronbach’s alpha α = 0.60, see Table 1). The combined means of these items were used for the analyses.

### 2.8. Psychosocial Factors

The psychosocial factors were measured as proposed in the RANAS model [18]. A description of the items can be found in Table A1 in Appendix B. Each factor was measured with at least one item. In cases where two or more items were used to measure a factor, the mean of these items was used for the analyses. The how-to-do knowledge for calling the Ebola Hotline was operationalized with a dichotomous item with responses coded as zero (did not know the number of the Ebola Hotline) or one (knew the number).

### 2.9. Statistical Analyses

Statistical analyses of the data were calculated with IBM SPSS 22 Statistics software (IBM SPSS Statistics for Windows, Version 22.0, IBM Corp., Armonk, NY, USA). Frequencies, multiple linear regression, and multiple mediation models were computed using the SPSS PROCESS macro [27]. Only psychosocial factors that were significant predictors within the multiple linear regression analyses were included in mediation models as mediators (M). Promotion activities were included as predictors (X) and EVD preventive behaviors as outcomes (Y) in parallel multiple mediator models. The specific indirect (a x b), direct (c’), and total effects (c) were calculated. A specific indirect effect is the effect of promotion activity via psychosocial factors on EVD preventive behaviors. The direct effect is the effect of the promotion activity on the EVD preventive behavior when the mediators (psychosocial factors) are not present in the model (X on Y independent of M). The total effect (c) is the sum of the specific indirect effect (a x b) and the direct effect (c’) (Appendix A). 

## 3. Results

In terms of gender, 434 (87.3%) of the respondents were female, and 63 (12.7%) were male. The reported age of the respondents ranged from 15 to 80 years (M = 35.89, SD = 13.22). On average, 12 people lived in the same household (SD = 7.68), which was defined as living in the same compound. The mean number of children under the age of five in the study households was three (SD = 1.92). 

### 3.1. Reach of the Promotional Activities

The analyses included four activities that promoted the preventive behaviors (see Table 2). The channel with the highest reach was the household visit, which reached 67% of respondents, followed by the poster at 63%. Nearly half of the respondents, 47%, received at least two items of the hygiene kit, and 39% of the respondents knew the Ebola information sheet. 

### 3.2. Psychosocial Factors Influencing Ebola Prevention Behaviors

Multiple regression analysis was conducted to reveal the influence of sociodemographic variables on the three behaviors (see Table A2 in Appendix B). However, the explained variance was very low, so these results were not taken into account.

On average, respondents stated that they wash their hands with soap and water at most key times. Handwashing after using the toilet was more frequently practiced than handwashing before eating. For the analysis, the two key times for handwashing with soap were combined (see Table 3). On average, the respondents said that they were willing and that they strongly intended to call the Ebola Hotline if there was an EVD case in the household. The same was found for the intention not to touch someone who might be suffering from EVD (see Table 3). However, the results showed that 89.6% of the respondents did not know the number of the Ebola Hotline.

### 3.3. Handwashing with Soap and Water

A multiple linear regression analysis was used to answer the first research question. The analysis revealed that six psychosocial factors significantly predicted the handwashing frequency (see Table 4). The model explains 48.5% of the variance in the self-reported handwashing frequency.

Conditional vulnerability (not protecting) (β = 0.149), which means thinking that the probability of an infection with EVD is high if they do not protect themselves with regular handwashing, was significantly associated with handwashing. Cost belief (costs) (β = 0.124), which means thinking that always washing hands with soap is expensive, was associated with increased handwashing, as was the Response belief (β = 0.123). This means the perceived certainty that always washing hands with soap and water prevents diseases like EVD or diarrhea. Furthermore, all three norm factors were significantly related with higher handwashing frequency: Others’ behavior (β = 0.305), which means the perception that other family members and people in the village wash hands with soap and water. Others’ (dis)approval (β = 0.123) means that people who are important to them at home or in the village approve of handwashing with soap and water and personal importance (β = 0.106) means the perception of handwashing as a personal obligation. 

### 3.4. Calling the Ebola Hotline

To answer the second research question, a multiple linear regression analysis was calculated. Four psychosocial factors were determined as significant predictors for the intention to call the Ebola Hotline and to report a suspected EVD case in the household (see Table 5). The model explained a variance of 27.3% in the intention to call the Ebola Hotline and report a suspected EVD case. A higher intention to call the Ebola Hotline was significantly related with Response belief (β = 0.195), which means study participants who think that calling the Ebola Hotline will help the person who might be suffering from Ebola. Then, others’ behavior in the household (β = 0.108), which means respondents who think that many people from their own household would call the Ebola Hotline, contributed significantly to explaining the intention to call the Ebola Hotline. Feeling committed to calling the Ebola Hotline (β = 0.226) was the most important predictor of the intention to call the Ebola Hotline. Communication (β = 0.133), which means that people who talk often about the Ebola Hotline are more likely to have a higher intention to call the Ebola Hotline than people who talk less often about it, was also a significant predictor. 

### 3.5. Not Touching a Person Who Might be Suffering from EVD

To answer the third research question, another multiple linear regression analysis was calculated. This regression analysis revealed that five psychosocial factors significantly predicted the intention not to touch someone who might be suffering from EVD (see Table 6). The psychosocial factors explained 17.1% of the variance of the intention not to touch someone who might be suffering from EVD.

A higher intention not to touch a sick person was significantly associated with respondents who have higher health knowledge about EVD (β = 0.101) and with respondents who think that they are at risk if they touch a sick person who might have EVD (β = 0.114). Response belief (β = 0.148), which means being certain that not touching a sick person who might have EVD prevents infection with EVD, was another significant predictor of the intention not to touch someone who might be suffering from EVD.

Furthermore, commitment to touch (β = 0.125) and commitment not to touch (β = 0.250) correlated with a higher intention not to touch someone who might be suffering from EVD.

### 3.6. Mediation Effects on Implemented Promotional Activities

To answer the fourth research question, a multiple mediation analysis was conducted. The aim was to reveal which interventions were significantly associated with handwashing and the reason for this relation. This was achieved by specifying the psychosocial factors through which the promotional activities addressed the preventive behaviors. Subsequently, it can be shown which of the crucial psychosocial factors were not tackled by the promotional activities. 

#### 3.6.1. Handwashing with Soap

All significant predictors from the regression analysis were selected to examine the indirect and direct effects of promotional activities on handwashing by means of mediation analysis. Table 7 presents the association of the EVD promotions with the key psychosocial factors and their specific indirect, direct, and total effects on handwashing. When looking at the total effects, three promotional activities were significantly associated with increased handwashing behavior: home visit, poster, and info sheet (see Table 7). The hygiene kit did not have a significant total effect on handwashing. The relation of the promotional activities with the key psychosocial factors can explain these associations. Poster and home visits were associated with all but one key psychosocial factor. The info sheet was associated with only two key psychosocial factors, and the hygiene kit was not associated with any of the key psychosocial factors and, thus, not with handwashing either. Furthermore, all of the key psychosocial factors were significantly associated with at least one of the promotional activities.

#### 3.6.2. Intention to Call the Ebola Hotline 

Again, all significant predictors from the regression analysis were integrated in a mediation analysis in order to examine the indirect and direct effects of promotional activities on the intention to call the Ebola Hotline. The hygiene kit was excluded from the analysis because most items of the kit did not include information about the Ebola Hotline. Table 8 presents the association of the EVD promotions with the key psychosocial factors and their specific indirect, direct, and total effects on handwashing. When looking at the total effects, no promotional activity was significantly associated with an increased intention to call the Ebola Hotline (see Table 8). Nevertheless, relations between the promotional activities and the key psychosocial factors were significant. The poster was associated with three key psychosocial factors, the home visit was associated with all key psychosocial factors, and the info sheet was associated with two of the key psychosocial factors. Again, all of the key psychosocial factors were significantly associated with at least one of the promotional activities. 

No mediation analyses were executed for not touching a sick person, for two reasons. The first was because of the low explanation by the psychosocial factors of the variance regarding the intention not to touch someone who might be suffering from EVD. The second was because not touching a sick person was not promoted in the same way as handwashing with soap and water or in the same way as calling the Ebola Hotline.

## 4. Discussion

This study aimed to evaluate the activities intended to promote Ebola prevention behaviors. However, it is not only crucial to know whether a promotional activity increased a behavior or not. It is equally crucial to understand the reasons for this effect. To achieve this, this study also examined the underlying psychosocial factors to identify which of them were affected by a promotional activity. 

### 4.1. Psychosocial Factors Influencing EVD Preventive Behaviors

In line with the findings of several knowledge-attitude-practice (KAP) surveys [5,6], we found that health knowledge about Ebola virus disease (EVD) was significantly associated with the intention not to touch someone who might be suffering from EVD. According to Bandura [28], individuals are more likely to adopt a new behavior if they have greater knowledge about the symptoms of a disease and about the prevention of the disease. In contrast, various other studies have found that factual knowledge is secondary to a range of other factors [14,15,29].

The findings about health knowledge relate to those about the response belief (perceived certainty that a behavior will prevent a disease), which was also a predictor of handwashing with soap and of the intentions to call the Ebola Hotline and not to touch a person who might be suffering from EVD. Jalloh et al. [7] found that handwashing with soap to avoid Ebola infection was mentioned unprompted by 66% of his sample. Response belief also explained stool-related handwashing in Haiti during the cholera outbreak [20]. However, in this study, response belief is not only the belief that a certain behavior prevents contracting EVD. It is also the belief that the public health infrastructure and system is able to handle the epidemic. Together with commitment, response belief was the most influential factor for the intention to call the Ebola Hotline. This belief is crucial for preventing the spread of a disease such as EVD. 

Our findings about health knowledge and response belief are in line with those about perceived vulnerability of the population. The respondents perceived the probability of contracting EVD as high if they did not protect themselves with regular handwashing with soap and water. Additionally, those respondents who thought that they were at risk if they touched a sick person who might have EVD were more likely to have a higher intention not to touch such a person. Jalloh et al. [7] found that the majority of respondents knew that avoiding contact with an infected corpse could prevent Ebola. Whether a perceived threat affects handwashing is consistent with previous research in emergency contexts. Curtis et al. [30] found in their review of motivational, planning, and habitual factors of handwashing in 11 countries that handwashing frequency increased during cholera epidemics (Uganda, Senegal, Kenya, and Peru) and sank again after outbreaks. 

Additional findings of our study were that respondents tended to wash hands more often than others did if they perceived that other people around them often wash hands with soap and water and believed that other people important to the respondents expect them to wash their hands. The more the respondents in this study perceived that many people from their own household would call the Ebola Hotline, the higher was their own intention to call the Ebola Hotline. Previous research has shown that norms are highly relevant to handwashing behavior [19,21,31] and other behaviors, including use of deep tube wells [32] and contraceptive methods [33].

The finding that the belief that always washing hands with soap is expensive is associated with handwashing might result from the experience of the respondents that soap has to be purchased more frequently when they perform this behavior continuously. 

Commitment was the most important predictor of the intention to call the Ebola Hotline and the intention not to touch a person who might be suffering from EVD. Commitments important for various WASH behaviors in developing countries has been shown by several previous studies [19,22,34,35].

In contrast, the factor commitment to touch explained the intention not to touch a person who might be suffering from EVD. Further analysis showed that 60 respondents, or 12%, of the study participants felt simultaneously committed to not touching and to touching someone who might be suffering from EVD. This might be explained by ambivalence and may be determined by culture and religion. If a respondent said they would not be willing to avoid touching a person who might be suffering from EVD, they were asked to say why not. The most commonly cited reasons included the need, the willingness, and the duty to help the sick person, and the fact that the sick person might be a family member, especially a close relative or a child. Similar to these findings, Lee-Kwan et al. [36] found that barriers to safe burials included the perception of bodies that were improperly handled and the fear that stigma may occur if a family member receives a safe, dignified medical burial. This may well explain the wavering between the commitments not to touch and to touch a person who might be suffering from EVD. Nevertheless, this fact could be crucial for preventing or curtailing the spread of EVD and should, therefore, be integrated in promotional activities.

The intention to call the Ebola Hotline was significantly related to communication, which means that people who talked often about the Ebola Hotline tended to have a higher intention to call the Ebola Hotline than others. One-to-one communication plays an essential role in a health behavior change, which has been confirmed by Rimal et al. [26] in their study about cardiovascular disease-related behaviors such as dieting, exercising, and smoking. Winters et al. [8] showed that, in the Ebola outbreak in Sierra Leone, the exposure to information sources was associated with higher knowledge and protective behaviors. 

Overall, the models were able to explain a substantial part of the variance of handwashing with soap (48.5%) and of the intention to call the Ebola Hotline (27.3%). Less variance was explained for the intention not to touch a person who might be suffering from EVD (17.1%), which means that we do not know clearly which psychosocial factors drove this.

### 4.2. Effects of Promotional Activities 

The three most effective promotions for handwashing were home visit, poster, and info sheet. Some studies have found positive effects of home visits [37,38], even though, in the study by Contzen and Mosler [20], home visits were negatively associated with handwashing behavior. A study from Thailand [39] found that posters were significantly and positively related with health knowledge but showed a tendency to be negatively related with handwashing behavior, and this was also the case in a study from Haiti [20]. In an analysis of a range of communication channels for promoting hygiene behavior, Pinfold [39] found that printed media such as stickers, posters, and leaflets were associated with significantly higher scores in health knowledge than other channels. However, this positive effect could not be found for behavior. Marais et al. [40] propose a multimethod communication in their eight-step model of health promotion. 

In our study, the hygiene kit did not have a significant association with handwashing, nor was it associated with any of the key psychosocial factors. Providing people with infrastructure alone and expecting the target health behavior to occur has been criticized by several authors [18,41,42]. 

Promotional activities are only successful when they target the key psychosocial factors. The results of this study suggest that all key psychosocial factors mediated the associations of the home visit, the poster, and the info sheet with handwashing. These three promotional activities were effective in tackling handwashing behavior because of the associations between the promotional activities and key psychosocial factors. The hygiene kit was not related to any of the key psychosocial factors and so could not address the behavior. In the main section, the handwashing promotional activities evaluated in this study were very successful in tackling the key psychosocial factors. 

The analysis indicated that the promotional activities for calling the Ebola Hotline were significantly related with all the relevant psychosocial factors. Nevertheless, none of the activities were associated with the intention to call the Ebola Hotline. This might be explained by the fact that calling the Ebola Hotline was not a focus of the promotional activities evaluated in this study. 

### 4.3. Practical Implications

The findings of this study can serve as a baseline for a further study of handwashing with soap and especially of the intention to call the Ebola Hotline. To change behavior successfully, promotion activities must target those factors that influence behavior. The findings of this study demonstrate that the norm factors, especially others’ behavior, response belief, and commitment, emerged as especially relevant to handwashing, the intention to call the Ebola Hotline, and not touching a person who might be suffering from EVD. The greater relevance of social norms and other factors than risk factors to health behaviors has been shown in a multi-country review about socio-psychological determinants for safe drinking water consumption behaviors [43].

We found that the home visit, the poster, and the info sheet were successful promotional activities in tackling handwashing behavior because they targeted the key psychosocial factors of handwashing behavior. The RANAS model provides behavior change techniques corresponding to psychosocial factors [44]. To increase handwashing behavior, the five psychosocial factors underlying handwashing with soap have to be tackled. The first of these, others’ (dis)approval, can be addressed by giving away stickers that bear a picture of an opinion leader washing his or her hands with soap and water. The second, personal importance, can also be addressed if the stickers mention that people in that household wash their hands with soap at key times and that they are good examples for others such as children. The third factor, others’ behavior, may be addressed by a community meeting to increase the perception of what others are doing and by providing participants with a commitment sign to hang up outside their houses. A health worker can inform the participants about their personal risk (conditional vulnerability not protecting) and, together with a doctor from the health facility—others’ (dis)approval—explain that handwashing with soap will protect them from EVD and diarrheal diseases (response belief).

For the intention to call the Ebola Hotline, five underlying psychosocial factors need to be targeted. In a radio advert, various kinds of people (others’ behavior household) could pledge their intention to call the Ebola Hotline if there is a suspected EVD case in their household (commitment). They believe that this service has to be used to help the affected person and to protect other family members and the members of their community (response belief). At the end of the advert, they ask: “And you, do you also commit yourself to calling the Ebola Hotline if there is a person who might be suffering from Ebola in your household?” (commitment and communication).

### 4.4. Limitations

The results have to be interpreted with caution because studies in an emergency context are especially prone to certain limitations. For ethical reasons, it is not appropriate to use a control group. Therefore, the present study was a cross-sectional study on the factors explaining EVD prevention behaviors and associations between promotional activities, psychosocial factors, and behavior or behavioral intention. However, no conclusions can be drawn about causality. 

One limitation is that the interventions implemented by the NGO attracted special attention from the population because of the Ebola outbreak in neighboring countries. Therefore, the impact of the interventions might be smaller when an outbreak is decreasing or even happened long ago. 

Another limitation is that we measured the intention of the Ebola-preventing behaviors in a country, which was not directly affected. However, the link between intention and behavior is strong (but not perfect), which has been shown by several studies [45]. Therefore, we can assume that the intention will result in Ebola preventing behaviors.

Measuring handwashing by self-report has been criticized by several scientists [46,47]. However, since the time for the survey was very limited, we could not directly observe handwashing behavior. Therefore, an over-reporting bias for the frequency of handwashing with soap is very likely. It would be useful to include further measurements as proxies [48,49] in the analysis. 

In the present study, we did not find relationships between sociodemographic data and behaviors. Seimetz et al. [21] found that self-reported handwashing was not explained by such factors as age, education level, or marital status. Other researchers have suggested that a higher education level and higher age are significantly related to self-reported handwashing frequencies [50,51]. Regarding wealth, studies have found economic status to be significantly associated with hand cleanliness [52], soap availability in the household, and observed handwashing behavior [53,54]. In contrast, Ram et al. [55] found in Senegal that none of their rapid handwashing measures were significantly related to observed handwashing behavior in models including wealth. The same was found in their studies in Peru and Vietnam. However, a comparison between studies is difficult. The study by Seimetz et al. [21] into the influence of contextual and psychosocial factors on handwashing did not find wealth to be a predictor of self-reported handwashing frequencies. This means that the psychosocial factors fully explained the effect of wealth on handwashing behavior. Therefore, the authors conclude that hygiene promotions should focus on psychosocial factors rather than sociodemographic factors [21]. 

The present study focused on psychosocial factors but enabled structural components of a behavioral change to take into account. Future studies should analyze the connection between physical, social, and personal context factors and psychosocial factors in a behavioral change. 

The fact that some of the respondents might have experienced several promotional activities and that some combinations might have another effect on behavior than others was not taken into account. Nevertheless, interaction effects should be considered in future studies. We did not integrate the attitudes of the respondents to the various promotional channels and activities in the analyses. Examining attributes of promotions such as its frequency, its likeability, its persuasiveness, and its trustworthiness may be important when evaluating a promotion channel or activity.

## 5. Conclusions

The present study demonstrates that some EVD prevention promotions were associated with the target behavior, and this was because they were associated with the key psychosocial factors steering the behavior. Conversely, promotions that were not associated with the behavior were not associated with the key psychosocial factors. The findings show the important role that psychosocial factors play in prevention behaviors during an EVD outbreak. Behavioral change programs should use evidence to target the right psychosocial factors and, thus, maximize their effects on prevention behaviors, especially in emergency contexts. Social norms and response beliefs were revealed as crucial for the prevention of EVD in the Gambia. However, the RANAS model used in this case focuses only on changes that can be achieved by individuals and households [18]. Changes at other levels, such as the institutional, political, and systemic, are often needed in order to control an outbreak of a contagious disease such as EVD and to influence people’s behavior. A situation such as that in West Africa during the last outbreak of EVD requires adequate public health infrastructure, public health resources, and corresponding and culturally appropriate risk communication and health promotion. Different languages and dialects, clear illustrations to include illiterate people, and aspects such as a strong tradition of oral communication and traditional beliefs also have to be considered in the communication [56].

## Figures and Tables

**Table 1 ijerph-16-02020-t001:** Questions to measure the intention to follow the two prevention behaviors.

Factor	Wording
Intention to call the Ebola Hotline	How strongly do you intend to call the National Ebola Hotline if you have a person with suspected Ebola in your household? Now we would like to ask you to imagine yourself in a certain situation. Suppose you have been at the market the whole day to sell vegetables. At the end of the day, you go home, and you find a member of your family who is vomiting, and the vomit contains blood, which could be a symptom of Ebola. In those circumstances, how willing would you be to call the Ebola Hotline and report the suspected Ebola case in your household?
Intention not to touch someone who might be suffering from EVD	How strongly do you intend to not touch a sick person who might suffer from Ebola in your household? Now we would like to ask you to imagine yourself in a certain situation. Suppose you have been at the market the whole day to sell vegetables. At the end of the day, you go home, and you find a member of your family who is vomiting, and the vomit contains blood, which can be a symptom for Ebola. In those circumstances, how willing would you be to not touch the sick person, which reduces the risk of contracting Ebola?

**Table 2 ijerph-16-02020-t002:** Overview of promotion activities and percentage of people who experienced the promotion.

Ebola Prevention Promotion	Description	%
Household visit	Main goal: Discuss the signs and symptoms of Ebola, the transmission routes, and hygiene behavior for Ebola prevention.	67%
Poster with information about EVD at public places	Main goal: Disseminate key messages how to protect yourself from Ebola (handwashing with soap and water), reporting the Ebola case to the Ebola Hotline, and the symptoms of Ebola (headache, vomiting, fever, joint pain, and bleeding).	63%
Ebola information sheet for the household	Main goal: Disseminate key messages how to protect yourself from Ebola (handwashing with soap and water), reporting the Ebola case to a health facility, and the symptoms of Ebola (headache, vomiting, fever, joint pain, and bleeding).	39%
Hygiene kits	Included soap, bleach, and material for a tippy tap, cups, a bucket, and a flyer about Ebola (only counted if someone received at least two items)	47%

**Table 3 ijerph-16-02020-t003:** Means (M) and standard deviations (SD) of handwashing, the intention to call the Ebola Hotline, and the intention not to touch someone who might be suffering from Ebola.

Key Time/Dependent Variable	N	M	SD
After using the toilet	495	4.51	0.74
Before eating	496	4.26	0.99
Combined handwashing variable	496	4.38	0.78
Calling the Ebola hotline	497	4.11	0.78
Not touching	491	4.12	0.94

**Table 4 ijerph-16-02020-t004:** Linear regression analysis for psychosocial factors explaining handwashing with soap and water.

Factor Group	Psychosocial Factors	M (SD)	*p*-Value	*β*
Risk factors	Community vulnerability	2.86 (1.51)	0.563	0.035
	Vulnerability	2.37 (1.48)	0.114	0.098
	Severity	4.37 (0.88)	0.134	0.069
	Health knowledge	18.93 (4.36)	0.746	0.013
	Conditional vulnerability (not protecting)	3.88 (1.33)	0.001	0.149 ***
	Conditional vulnerability (protecting)	2.11 (1.26)	0.535	−0.023
Attitude factors	Cost belief (effort)	1.27 (0.73)	0.051	−0.087
	Cost belief (time)	1.41 (0.86)	0.782	−0.013
	Cost belief (costs)	2.44 (1.19)	0.004	0.124 **
	Cost belief (distance)	1.61 (0.99)	0.586	−0.022
	Feelings (like)	4.40 (0.71)	0.425	0.042
	Response belief	4.21 (0.89)	0.008	0.123 **
Norm factors	Others’ behavior	4.12 (0.76)	0.000	0.305 ***
	Others’(dis)approval	4.26 (0.67)	0.007	0.123 **
	Personal importance	4.31 (0.65)	0.041	0.106 *
Ability factors	How-to-do knowledge	3.69 (0.70)	0.075	0.074
	Confidence in performance	4.31 (0.82)	0.259	0.058
	Confidence in performance (water)	1.77 (1.16)	0.893	−0.006
	Confidence in performance (soap)	2.48 (1.20)	0.885	−0.007
	Confidence in performance (time)	1.49 (1.01)	0.923	−0.005
	Confidence in performance (distance)	4.07 (0.89)	0.174	−0.068
Self-regulation factors	Action planning	4.03 (0.67)	0.769	0.013
	Remembering	2.38 (1.41)	0.255	0.050
	Commitment	3.11 (0.83)	0.106	0.088

*Note:* * *p* ≤ 0.05, ** *p* ≤ 0.01, *** *p* ≤ 0.001. Adjusted R^2^ = 0.485. *N* = 422.

**Table 5 ijerph-16-02020-t005:** Linear regression analysis for psychosocial factors explaining the intention to call the Ebola Hotline and report a suspected EVD case.

Factor Group	Psychosocial Factors	M (SD)	*p*-Value	*β*
Risk factors	Community vulnerability	2.76 (1.52)	0.191	0.089
	Vulnerability	2.30 (1.44)	0.749	−0.022
	Severity	4.30 (0.92)	0.400	−0.040
	Health knowledge	18.87 (4.43)	0.113	0.070
Attitude factor	Response belief	4.18 (0.85)	0.000	0.195 ***
Norm factors	Others’ behavior household	4.10 (1.18)	0.021	0.108 *
	Others’ (dis)approval household	4.28 (0.60)	0.630	0.027
	Others’ (dis)approval village	4.32 (0.57)	0.329	0.047
	Personal importance	4.21 (0.64)	0.060	0.096
Ability factors	How-to-do knowledge	n. a.	0.845	0.008
	Confidence in performance	4.20 (0.84)	0.751	−0.014
Self-regulation factors	Commitment	4.45 (0.61)	0.000	0.226 ***
Additional factor	Communication	3.06 (1.38)	0.002	0.133 **

*Note*. * *p* ≤ 0.05, ** *p* ≤ 0.01, *** *p* ≤ 0.001. Adjusted *R^2^* = 0.273. *N* = 467.

**Table 6 ijerph-16-02020-t006:** Linear regression analysis for psychosocial factors explaining the intention to not touch someone who might be suffering from EVD.

Factor Group	Psychosocial Factors	M (SD)	*p*-Value	*β*
Risk factors	Community vulnerability	2.76 (1.51)	0.242	0.084
	Perceived vulnerability	2.28 (1.44)	0.875	−0.011
	Perceived severity	4.29 (0.92)	0.719	−0.019
	Health knowledge	19.01 (4.30)	0.029	0.101 *
	Conditional vulnerability touching	4.30 (0.99)	0.020	0.114 *
Attitude factor	Response belief	4.22 (0.85)	0.002	0.148 **
Self-regulation factors	Control not to touch	4.27 (0.71)	0.686	0.019
	Commitment to touch	1.89 (1.26)	0.019	0.125 *
	Commitment not touch	4.22 (0.83)	0.000	0.250 ***

Note: * *p* ≤ 0.05, ** *p* ≤ 0.01, *** *p* ≤ 0.001. Adjusted *R^2^* = 0.171. *N* = 467.

**Table 7 ijerph-16-02020-t007:** Mediation analysis: effects of promotional activities on self-reported handwashing via psychosocial factors (mediators).

Promotional Activity	(a)	Psychosocial Factors/Mediators	(b)	Specific Indirect Effect (a*b) 95% CL [LL, UL]	Direct Effect (c’)	Total Effect (c)
Poster	0.55 *** (0.000)	Cond. vulnerability not protecting	0.13 *** (0.000)	0.07, [0.03, 0.12]		
	0.22 (0.063)	Cost belief	0.03 (0.241)	0.01, [−0.00, 0.03]		
	0.37 *** (0.000)	Response belief	0.14 *** (0.000)	0.05, [0.02, 0.11]		
	0.64 *** (0.000)	Others’ behavior	0.31 *** (0.000)	0.20, [0.12, 0.29]		
	0.19 ** (0.007)	Others’ (dis)approval	0.10 * (0.041)	0.02, [0.00, 0.05]		
	0.21 ** (0.002)	Personal importance	0.14 ** (0.008)	0.30, [0.01, 0.07]		
					0.20 *** (0.001)	0.57 *** (0.000)
Home visit	0.62 *** (0.000)	Cond. vulnerability not protecting	0.12 *** (0.000)	0.07, [0.04, 0.12]		
	0.05 (0.708)	Cost belief	0.04 (0.108)	0.001, [−0.01, 0.02]		
	0.48 *** (0.000)	Response belief	0.13 *** (0.000)	0.06, [0.02, 0.12]		
	0.69 *** (0.000)	Others’ behavior	0.30 *** (0.000)	0.21, [0.13, 0.31]		
	0.27 *** (0.000)	Others’ (dis)approval	0.10 * (0.037)	0.03, [0.01, 0.07]		
	0.33 *** (0.000)	Personal importance	0.15 ** (0.004)	0.05, [0.01, 0.11]		
					0.23 *** (0.000)	0.65 *** (0.000)
Info sheet	0.00 (0.999)	Cond. vulnerability not protecting	0.13 *** (0.000)	0.00, [−0.04, 0.03]		
	0.54 *** (0.000)	Cost belief	0.03 (0.230)	0.02, [−0.01, 0.04]		
	0.02 (0.784)	Response belief	0.14 *** (0.000)	0.00, [−0.02, 0.03]		
	0.39 *** (0.000)	Others’ behavior	0.34 *** (0.000)	0.14, [0.07, 0.21]		
	0.09 (0.205)	Others’ (dis)approval	0.09 (0.062)	−0.01, [−0.03, 0.01]		
	0.01 (0.841)	Personal importance	0.16 ** (0.003)	0.00, [−0.02, 0.03]		
					0.01 (0.830)	0.16 * (0.031)
Hygiene kit	0.04 (0.865)	Cond. vulnerability not protecting	0.09 ** (0.001)	0.01, [−0.05,0.07]		
	−0.15 (0.550)	Cost belief	−0.02 (0.469)	0.00, [−0.01,0.03]		
	0.03 (0.845)	Response belief	0.20 *** (0.000)	0.01, [−0.06,0.09]		
	0.11 (0.408)	Others’ behaviour	0.36 *** (0.000)	0.04, [−0.05,0.13]		
	0.01 (0.948)	Others’ (dis)approval	0.03 (0.606)	0.00, [−0.02,0.02]		
	0.20 (0.097)	Personal importance	0.18 ** (0.004)	0.04, [−0.01,0.13]		
					0.01 (0.947)	0.09 (0.486)

Note: * *p* ≤ 0.05, ** *p* ≤ 0.01, *** *p* ≤ 0.001. Displayed are unstandardized betas and p-values. Poster: *N* = 401, *R^2^* = 0.50 (b), home visit: *N* = 400, *R^2^* = 0.49 (b), info sheet: *N* = 401, *R^2^* = 0.47 (b), hygiene Kit: *N* = 288, *R^2^* = 0.42. Number of bootstrap samples for bias-corrected bootstrap confidence intervals: 10,000. Level of confidence for all confidence intervals: 95%.

**Table 8 ijerph-16-02020-t008:** Mediation analysis: effects of promotional activities on calling the Ebola Hotline via psychosocial factors (mediators).

Promotional Activity	(a)	Psychosocial Factors/Mediators	(b)	Specific Indirect Effect (a*b) 95% CL [LL, UL]	Direct Effect (c’)	Total Effect (c)
Poster	0.19 * (0.036)	Response belief	0.23 *** (0.000)	0.04, [0.01, 0.10]		
	0.16 (0.185)	Others’ behavior household level	0.07 ** (0.019)	0.01, [−0.00,0.04]		
	0.26 *** (0.000)	Commitment	0.34 *** (0.000)	0.09, [0.04, 0.17]		
	0.46 *** (0.001)	Communication	0.08 *** (0.003)	0.03, [0.01, 0.07]		
					−0.13 (0.083)	0.05 (0.542)
Home visit	0.22 * (0.016)	Response belief	0.18 *** (0.000)	0.04, [0.01, 0.09]		
	0.31 * (0.014)	Others’ behavior household level	0.08 * (0.013)	0.02, [0.01, 0.06]		
	0.38 *** (0.000)	Commitment	0.40 *** (0.000)	0.16, [0.09, 0.25]		
	0.54 *** (0.000)	Communication	0.08 *** (0.001)	0.04, [0.01, 0.09]		
					−0.15 * (0.047)	0.11 (0.200)
Info sheet	−0.09 (0.294)	Response belief	0.23 *** (0.000)	−0.02, [−0.07, 0.02]		
	0.30 * (0.012)	Others’ behavior household level	0.08 * (0.013)	0.02, [0.01, 0.06]		
	0.04 (0.518)	Commitment	0.31 *** (0.000)	0.01, [−0.02, 0.06]		
	0.95 *** (0.000)	Communication	0.06 * (0.017)	0.06, [0.01, 0.12]		
					0.02 (0.796)	0.09 (0.244)

Note: * *p* ≤ 0.05, ** *p* ≤ 0.01, *** *p* ≤ 0.001. Displayed are unstandardized betas and *p*-values. Poster: N = 395, *R^2^* = 0.27 (b). Home visit: N= 394, *R^2^* = 0.27 (b). Info sheet: *N* = 396, *R^2^* = 0.25 (b). Number of bootstrap samples for bias-corrected bootstrap confidence intervals: 10,000. Level of confidence for all confidence intervals: 95%.

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
