# Peer review of "The Impact of Various Promotional Activities on Ebola Prevention Behaviors and Psychosocial Factors Predicting Ebola Prevention Behaviors in the Gambia Evaluation of Ebola Prevention Promotions"

_ijerph, 2019, doi:10.3390/ijerph16112020_

Round 1

Reviewer 1 Report

This was a thorough and well presented paper describing interventions to influence behaviour to break the transmission link in highly infectious diseases.  Although focused on EVD it borrowed from experiences with other diseases and therefore also has wider relevance for other infections.

The factors and interventions explored; handwashing, not touching an infected person and calling for help are vitally important in tackling disease outbreaks. the conclusions and observations are useful and thought provoking.

I have a few minor editorial corrections and some observations to consider.

Line 87 refers to 'tippy tap'.  I was familiar with this but not all readers will be.  I suggest you add in a link to information - www.tippytap.org/the-tippy-tap.

In Table 1 second part the scenario is posed about not touching a family member with infectious disease symptoms.  While the response obviously should be not to, this will be strongly conflicted with empathetic human nature to help especially a family member.  Was this conflict, how it made the interviewee feel, and how to resolve it discussed during the interviews? I note that this is explored in lines 406 to 415 in Discussion but if more could be said it would be interesting.

Line 230 states respondent ages. Unless Gambia is different from Sierra Leone, stated ages on medical forms there during the outbreak were obviously in some cases estimates.  It might be more accurate to describe it as 'stated' or 'reported' ages?

While not disputing the results Table 2 that information sheets for households were much less effective than posters in public places I found it surprising. As it was observed but not explored in the Discussion, to inform future interventions do the authors have any views as to why information sheets in households were less effective and any ways to improve impact?   

Line 394 didn't make sense to me - if I have got the meaning correct I suggest reword as '...soap and water, and believed it was important that others expect them to wash their hands, tended....' 

I note in Appendix A under Abilities and Self-regulation the acknowledged limiting factors to hand washing ie ready availability of water in the household and planning to ensure there is.

For future work, having emphasised the importance of hand washing whether there is any value in exploring education in effective handwashing ie ensuring all parts of the hands are rubbed.

Author Response

Response to Reviewer 1:

Line 87 refers to 'tippy tap'.  I was familiar with this but not all readers will be.I suggest you add in a link to information - www.tippytap.org/the-tippy-tap.

·        We added a link www.tippytap.org/the-tippy-tap, as requested (line 86).

In Table 1 second part the scenario is posed about not touching a family member with infectious disease symptoms.  While the response obviously should be not to, this will be strongly conflicted with empathetic human nature to help especially a family member.  Was this conflict, how it made the interviewee feel, and how to resolve it discussed during the interviews? I note that this is explored in lines 406 to 415 in Discussion but if more could be said it would be interesting.

·        We addressed this comment and added a sentence regarding this issue (lines 402-404).

Line 230 states respondent ages. Unless Gambia is different from Sierra Leone, stated ages on medical forms there during the outbreak were obviously in some cases estimates.  It might be more accurate to describe it as 'stated' or 'reported' ages?

·        We changed the wording (line 224).

While not disputing the results Table 2 that information sheets for households were much less effective than posters in public places I found it surprising. As it was observed but not explored in the Discussion, to inform future interventions do the authors have any views as to why information sheets in households were less effective and any ways to improve impact?

·        We assume that a poster gains more attention from people living in a community than an information sheet which may disappear after a short time in the household. While reading a poster and seeing others reading a poster about transmission routes of a contagious disease such as Ebola virus disease tackles not only the factor others’ behaviour within the family and the household (what an information sheet for a household does) but also the norm or others’ behaviour from the whole community. To improve impact, an information sheet could be developed together with a person who is important for the community and who has the power to persuade the people living there.

Line 394 didn't make sense to me - if I have got the meaning correct I suggest reword as '...soap and water, and believed it was important that others expect them to wash their hands, tended....' 

·        We reworded the sentence (line 385).

I note in Appendix A under Abilities and Self-regulation the acknowledged limiting factors to hand washing ie ready availability of water in the household and planning to ensure there is.

For future work, having emphasized the importance of hand washing whether there is any value in exploring education in effective handwashing ie ensuring all parts of the hands are rubbed.

Reviewer 2 Report

The study evaluated the effectiveness of EVD prevention promotions in Gambia and assessd the related psychosocial factors. The study is conducted well and the results are comprehensive. Some minor comments:

1. How the authors determined the sample size?

2. Please report the p-values instead of using ** as there are a number of tests.

3. The explanation of mediation analysis is unclear.

4. The format of Table 7 and 8 should be refined as it is hard to read.

5. English editing is required.

6. There are too many results and the authors should summarize all points clear.

Author Response

Response to Reviewer 2:

1. How the authors determined the sample size?

·        Sample size estimation with G*Power 3.1 yielded a total sample size of 400 households to detect a small to medium effect in Cohen’s f 2 at the Type I error probability of 0.05 and a statistical power of 0.95. Due to the normal drop-out rate, we aimed to collect data from 500 households.

2. Please report the p-values instead of using ** as there are a number of tests.

·        The p-values express the level of significance in one test of one regression or one mediation model. They are specified below the tables in notes (*p ≤ .05, **p ≤ .01, ***p ≤ .001.)

3. The explanation of mediation analysis is unclear.

·        We included an explanation (lines 212-221).

4. The format of Table 7 and 8 should be refined as it is hard to read.

·        We added an explanation of mediation analysis in methods section (see Figure 1, page 18).

5. English editing is required.

·        Our manuscript was edited by professional language services (Dr. Simon Milligan, ETH and University of Zurich, milliganediting@gmail.com).

6. There are too many results and the authors should summarize all points clear.

·        We summarized findings in the practical implications section (lines 448-477)

Reviewer 3 Report

Lines 103 and 104 should be rewritten.

Section 2.9 is very confusing, especially due to Line 224.

Author Response

Response to Reviewer 3:

Lines 103 and 104 should be rewritten.

·        These lines were rewritten (99-105).

Section 2.9 is very confusing, especially due to Line 224.

·        We did requested changes in the section 2.9 (explanation of mediation analysis, lines 212-221).

Round 2

Reviewer 2 Report

The authors require to report EXACT p-values instead of a range e.g. 'p<0.05'. Many statistical reporting guidelines have indicated this problem. Except this, I have no further comments.

Author Response

Dear Reviewer

The authors require to report EXACT p-values instead of a range e.g. 'p<0.05'. Many statistical reporting guidelines have indicated this problem. Except this, I have no further comments.

·        We added p-values as requested.